Landscape predictors influencing livestock depredation by leopards in and around Annapurna Conservation Area, Nepal

Lamichhane Saurav 1
Bhattarai Divya 2
Maraseni Tek 3
Shaney Kyle J. 4
Karki Jhamak Bahadur 5
Adhikari Binaya 6 10
Pandeya Pratik 1
Shrestha Bikram 7 8
Adhikari Hari hari.adhikari@helsinki.fi 9
1 Faculty of Forestry, Agriculture and Forestry University , Hetauda , Nepal
2 Institute of Botany and Landscape Ecology, University of Greifswald , Greifswald , Germany
3 Centre for Sustainable Agricultural Systems (CSAS), University of Southern Queensland , Toowoomba , Australia
4 Department of Biological and Health Sciences, Texas A&M University-Kingsville , Kingsville , Texas , USA
5 Kathmandu Forestry College, Tribhuwan University , Kathmandu , Nepal
6 Institute of Forestry, Pokhara Campus, Tribhuvan University , Pokhara , Nepal
7 Department of Biodiversity Research, Global Change Research Institute, Czech Academy of Sciences , Brno , Czech Republic
8 Green Governance Nepal (GGN) , Kathmandu , Nepal
9 Department of Geosciences and Geography, University of Helsinki , Helsinki , Finland
10 Current affiliation:  Department of Biology, University of Kentucky , Lexington , KY , USA
Caravaggi Anthony
Electronic publication date: 2023 Dec 13
Publication date: 2023
Volume: 11
Electronic Location ID: e16516
Received 2023 Jul 26; Accepted 2023 Nov 3
Copyright: ©2023 Lamichhane et al.
Copyright year: 2023
Copyright holder: Lamichhane et al.
License: This is an open access article distributed under the terms of the Creative Commons Attribution License, which permits unrestricted use, distribution, reproduction and adaptation in any medium and for any purpose provided that it is properly attributed. For attribution, the original author(s), title, publication source (PeerJ) and either DOI or URL of the article must be cited.
License URL: https://creativecommons.org/licenses/by/4.0/

Keywords: Livestock depredation, Annapurna Conservation Area, Landscape predictors

Funding: Rufford Foundation, UK 32618-1 The University of Helsinki The Ministry of Education, Youth, and Sports of CR within the CzeCOS program LM2023048 This work was supported by the Rufford Foundation, UK (32618-1), financially for this research. Open access funding was provided by the University of Helsinki. Bikram Shrestha was supported by the Ministry of Education, Youth, and Sports of CR within the CzeCOS program, grant number LM2023048. The funders had no role in study design, data collection and analysis, decision to publish, or preparation of the manuscript.

==============================
Livestock depredation by leopards is a pervasive issue across many Asian and African range countries, particularly in and around protected areas. Developing effective conflict mitigation strategies requires understanding the landscape features influencing livestock depredation. In this study, we investigated predictors associated with livestock depredation by leopards using 274 cases of leopard attacks on livestock that occurred between 2017 and 2020 in the Annapurna Conservation Area, Nepal. We also examined how livestock predation by leopards varied depending on the species, season, and time. A generalized linear model with binary logistic regression was used to test the statistical significance of variables associated with the presence and absence of conflict sites. The results revealed that the area of forest, agricultural land, length of rivers, slope, proximity to settlements and protected areas, and elevation significantly predicted the probability of leopard attacks on livestock. We also observed a significant increase in the incidence of leopard predation on livestock with decreasing slopes and rising elevations. The areas near human settlements and the protected areas faced a higher risk of leopard predation. The incidence of leopard predation on livestock varied significantly depending on the livestock species, season, and time. Goats were the most highly predated livestock, followed by sheep, cow/ox, and buffalo. A total of 289.11 km2 (or around 5% of the research area) was deemed to be at high risk for leopard predation on livestock. This study’s comprehensive understanding of human-leopard conflicts provides valuable insights for planning and implementing measures to reduce damage caused by leopard populations throughout their range.

Introduction

Addressing the global challenge of human-wildlife conflict is an urgent issue faced by managers and policymakers (Sharma et al., 2020; Torres, Oliveira & Alves, 2018). This conflict arises from the shared use of resources by humans, livestock, and wild predators (Venumière-Lefebvre, Breck & Crooks, 2022; Shrestha et al., 2022; Graham, Beckerman & Thirgood, 2005; Treves & Karanth, 2003). In cases of human-large carnivore conflict, animals kill livestock (Dalerum, Selby & Pirk, 2020; Lamichhane et al., 2018) and occasionally pose a threat to humans (Inskip & Zimmermann, 2009; Woodroffe et al., 2007), leading to their persecution by people (Mateo-Tomás et al., 2012). Carnivore habitats are increasingly fragmented and resource-scarce due to human population growth, anthropogenic activity reduced wild prey, and changing land use, resulting in human-carnivore conflicts (Dheer et al., 2022; Naha et al., 2020; Acharya et al., 2017; Inskip & Zimmermann, 2009). For instance, rapid human population growth and habitat encroachment on common leopard (Panthera pardus, referred to as “leopard” hereafter) habitats have led to reduced prey availability and habitat fragmentation. Those pressures drive leopards to forage closer to human settlements, leading to frequent human-leopard conflicts (Puri et al., 2020; Yadav et al., 2020; Stein et al., 2016).

The leopard is categorized as Vulnerable by the International Union for Conservation of Nature and Natural Resources (IUCN) Red List (Stein et al., 2016) and listed as Appendix 1 in CITES law. The National Red List of Mammals categorizes it as an endangered species. The legal status of this species in Nepal is Protected (Appendix 1) under the National Parks and Wildlife Conservation Act 1973 (Jnawali et al., 2011). Its populations are declining throughout most of their range (Jacobson et al., 2016; Athreya et al., 2011), leading to isolation, with some Asian subspecies now assessed as endangered and critically endangered on the IUCN Red List. They are widely distributed across Africa and Asia, but their historical range has seen a significant 61% reduction (Stein et al., 2016). The leopard is distributed in Nepal from the lowland Terai region to the mid-hill areas (Baral et al., 2023; Koirala et al., 2012), and recent observations indicate sightings at elevations of up to 4,500 meters in Annapurna Conservation Area (Bikram Shrestha, 2023, pers. comm.). However, their population status and occupancy are poorly understood (Lamichhane et al., 2021).

Numerous studies have been conducted to examine the spatio-temporal patterns and frequency of leopard predation on livestock in Nepal (Kandel et al., 2023; Adhikari et al., 2022; Dhungana et al., 2019; Lamichhane et al., 2018; Karki & Rawat, 2014; Koirala et al., 2012; Sijapati et al., 2021). The Department of National Parks and Wildlife Conservation (DNPWC, 2017) reported that leopards were responsible for 78% of livestock kill incidents in Nepal. Furthermore, Adhikari et al. (2022) identified the mid-hills regions as the areas with the highest risk for leopard predation on livestock. Likewise, similar studies have been conducted outside of Nepal (Akrim et al., 2021; Naha, Sathyakumar & Rawat, 2018; Shehzad et al., 2015; Constant, Bell & Hill, 2015; Qamar et al., 2010).

Few studies have been conducted on the influential factors affecting leopard predation on livestock (Naha et al., 2020; Yadav et al., 2020; Rostro-García et al., 2016; Constant, Bell & Hill, 2015) and mapping conflict hotspots (Yadav et al., 2020; Sharma et al., 2020). However, all of these studies were conducted outside of Nepal. So far, studies that employ a robust sampling and modelling framework have not been conducted in Nepal. Identifying zones prone to leopard predation on livestock is crucial for efficient conflict mitigation and leopard conservation. Focusing resources and efforts on these areas allows for targeted livestock protection, reduces retaliatory killings of leopards, and engages communities in sustainable coexistence strategies (Miller, 2015). This approach addresses the complex challenges of human-leopard conflicts while safeguarding both local livelihoods and leopard populations (Adhikari et al., 2022; Yadav et al., 2020; Mateo-Tomás et al., 2012). Reckoning this fact, we have chosen the Gandaki Province of Nepal as our research site for these unexplored studies, particularly as it overlaps with the Annapurna Conservation Area. This region, situated in the mid-hills of Nepal, is a notable focal point for human-leopard conflict (Adhikari et al., 2022; Koirala et al., 2012).

To address the aforementioned research gaps, we aim to determine important landscape variables that can influence leopard predation on livestock by employing a binomial generalized linear model and map high-conflict-prone areas by utilizing significant environmental, anthropogenic, and topographic variables through MaxEnt modeling. In this study, we established hypotheses to investigate how different landscape factors influence leopard predation on livestock in Nepal. The study’s primary objective was empirically assessing and confirming these hypotheses through data analysis and statistical methods. In addition, we conducted descriptive analyses to understand the spatio-temporal and predation patterns of leopards on livestock. The study’s results are expected to enhance efforts to reduce human-leopard conflicts and promote coexistence between humans and leopards.

Methodology

Study area

The Annapurna Conservation Area (ACA), established in 1992, is Nepal’s largest protected area, covering 7,629 sq. km. It is located in the hills and mountains of west-central Nepal (83057′E, 28050′N) and is managed by the National Trust for Nature Conservation (NTNC). The Annapurna Himalayas have various ecosystems, from subtropical woods to trans-Himalayan freezing deserts. The ACA is rich in biodiversity, home to 1,352 plant species, 128 wild mammal species, 514 bird species, 348 butterfly species, 40 reptile species, and 23 amphibian species (NTNC National Trust for Nature Conservation, 2018), and ranges in altitude from 790 m to the peak of Annapurna I at 8,091 m. It is the only protected area in Nepal where locals can live within the boundaries, own their private land, and keep their traditional rights and access to the usage of natural resources. The area is also a popular trekking destination for visitors from all over the world. To ease its management, ACA is divided into seven-unit conservation offices: Ghandruk, Lwang, Sikles, Bhujung, Manang, Jomsom and Lo-Manthang. The local community members reside in 15 rural municipalities of five districts and 87 wards. This study was conducted in three of the five Gandaki province districts (Kaski, Lamjhung, and Myagdi), which overlap with the ACA (Fig. 1). Myagdi and Lamjung districts cover relatively smaller parts of the conservation area (approximately 413.46 km2 and 386.41 km2), and Kaski district covers the largest part of the conservation area (1485.6 km2). The forest area outside the conservation area is under the jurisdiction of a division of forest offices. The Department of National Park and Wildlife Conservation (DNPWC) (077/78 Eco 45) and the National Trust for Nature Conservation, Annapurna Conservation Area Project (279/077/078) provided a research permit.

Figure 1 Study area boundary, showing the land cover and conservation area.

Data collection

The Government of Nepal (GoN) developed relief guidelines to assist local people with wildlife damage, including the leopard, in 2009 (Acharya et al., 2016). We compiled compensation data for HLC (leopard attacks on livestock) between 2017 and 2020 from the Divisional Forest Offices (DFO) in Myagdi, Kaski, and Lamjung and Conservation Area Management Units (CAMU) in Bujhung, Sikles, Lwang, and Ghandruk. The data were managed following the Nepalese Calendar, which runs from mid-July to mid-July (Bikram Sambat), and we used fiscal years to ensure data consistency. Although there were cases of leopards killing dogs and hens in the study area, we only recorded species for which compensation had been allocated to livestock owners by the government. Officials from the DFO and CAMU verified all reported data before compensation. Conflict information was gathered from 274 locations (“conflict sites” hereafter). In the study districts, we visited 252 conflict sites with elevations lower than 3,000 m. For the 22 conflict sites with elevations higher than 3,000 m, we used Google Earth Pro 7.3.3 and took help from local herders who were present during the conflict incident at the sites to identify the grid cell of the conflict site. The research team conducted field visits accompanied by staff from the DFO and CAMU of a particular village. The survey was conducted only after ethical approval from the DFO offices (Permission number: 1299) in each district and the CAMU offices inside conservation areas (Permission number: 279/077/078).

We located the owners who lost their livestock between 2017 and 2020 based on the information available from the compensation records. With the help of the livestock owners, we visited conflict sites. We recorded the GPS location of each site, along with information on the particular season and time of the incidents.

Data processing

Considering the limited resources, we had to maintain effectiveness by reducing the logistical complexity of the survey; hence, we stratified the study area into 4 km × 4 km (i.e., 16 km2) grids (n = 478) using ARC GIS 10.5 (ESRI, 2017). Out of 476 sampling grids, we recorded conflicts from 93 grids. The cell size was selected as 4 km × 4 km to reduce chances of autocorrelation and identify landscape drivers of human–leopard conflicts. Studies (Naha, Sathyakumar & Rawat, 2018; Naha et al., 2020) have used 2 km × 2 km–5 km × 5 km for the leopard conflict survey in previous studies. The grid size (4 km × 4 km) was selected based on the topographic features of the study area, to increase survey precision, alongside decreasing survey cost and logistical complexity.

We examined various potential explanatory variables by utilizing publicly available data layers previously explored in human-leopard conflict studies (Naha et al., 2020; Ramesh et al., 2020; Sharma et al., 2020; Yadav et al., 2020; Broekhuis, Cushman & Elliot, 2017; Rostro-García et al., 2016; Treves et al., 2011). We compiled a comprehensive set of 14 landscape predictors for each grid cell, categorizing them into five distinct groups (Table 1).

Table 1 Major predictor variables considered for spatial analysis in the project sites.

Types of variables	Predictor variable	Abbreviation	Unit	Range	Source	
Habitat variables	Area of agricultural land
Area of bare ground
Area under forests
Area of grassland
Area of shrubland	AAL
ABG
AF
AGL
ASL	m2
m2
m2
m2
m2	0–3,177,300
0–12,706,300
0–15,947,900
0–22,14,600
0–15,072,200	ESRI (2020)	
Protected area	Distance from the protected area	DPA	meter (m)	0–42,554.9		
Water	Area of water bodies
River length
Distance to water bodies	AWB
RL
DWB	m2
meter (m)
meter (m)	0–2,974,900
0–14,483.10
0–8,594.57	ESRI (2020)
OCHA Nepal (2021a)
OCHA Nepal (2021a)	
Human influence and infrastructure	Length of road
Distance from road
Distance from settlement	LR
DR
DS	meter (m)
meter (m)
meter (m)	0–107,248.5	OCHA Nepal (2021b)
OCHA Nepal (2021c)	
Topography	Slope
Elevation		(°)
meter(m)	2.15–51.9
415.1–7,420.4	ASF, 2021	

We accessed land cover data specific to Nepal from ESRI (2020) to generate land use types within our study area. Subsequently, for each grid cell, we computed the areas corresponding to various land cover classes, including agricultural land, forests, bare ground, grassland, and shrubland. We applied the “Spatial join” tool in Arc GIS 10.5. We relied on Nepal’s land cover data to assess water bodies within our study area. We extracted information regarding the river network from OCHA Nepal ( OCHA Nepal, 2021a) and computed the lengths of rivers within each grid cell through the ‘Intersect’ function in Arc GIS. To determine the distance from water bodies to each grid cell, we employed the “Euclidean distance” tool.

Regarding roads and settlements, we obtained shape files representing Nepal’s road network and settlements from OCHA Nepal ( OCHA Nepal, 2021b; OCHA Nepal, 2021c). Using the Euclidean distance tool, we then calculated the distances from roads and settlements to each grid cell. Additionally, we assessed the lengths of roads within each grid cell using the ‘Intersect’ function in Arc GIS. We derived mean slope and elevation values for each grid from the Digital Elevation Model (DEM) file available on the Alaska Satellite Facility (ASF) website (ASF, 2021). This process was facilitated by the ‘zonal statistics’ tool in ARC GIS. The nearest distance from protected areas boundary to the center of each grid cell was calculated for each grid using the Euclidean distance tool in Arc GIS. The points inside the protected area boundary were manually set as zero, the distance of points outside the boundary were calculated, and the datasets were merged to get the final distance value.

Data analysis

We also tested three additional temporal predictor variables for a relationship with leopard depredations, including time of day, month, and season. First, attacks were assigned to a time frame. We divided 24 h into four-hour intervals since the precise timing of livestock predation was unknown (12 AM–4 AM, 4 AM–8 AM, 8 AM–12 PM, 12 PM–4 PM, 4 PM–8 PM, 8 PM–12 AM). We also divided 12 months into three seasons of 4 months each (winter i.e., November–February, summer, i.e., March–June, monsoon, i.e., July–October). Statistical data analysis was done in the R Statistical package v 4.0.4 (R Core Team, 2021). We also used a chi-square (χ2) test to examine seasonal, monthly, and temporal variations in livestock depredation. To uncover more information about the features of kill sites, we divided the assigned predictors into groups (Table 2). Descriptive summaries based on month, season, and time of leopard attacks on livestock were calculated using the Pivot table function in Microsoft Excel 2013.

Table 2 Association of conflict sites within different landscape variables.

Variables	Classes	Frequency of attack (%)	
Slope (°)	2–10
10–20
20–30
30–40
>40	22
30
29
13
6	
Elevation (m)	415–1,000
1,000–2,000
2,000–3,000
>3,000	28
34
27
11	
Location	Outside conservation areas
Inside conservation areas	49
51	
Distance from road (m)	0–500
500–1,000
1,000–1,500
1,500–2,000
2,000–2,500
>2,500	44
23
14
6
4
9	
Distance from river (m)	0–500
500–1,000
1,000–1,500
1,500–2,000
2,000–2,500
>2,500	42
29
17
9
2
1	
Distance from settlement (m)	0–500
500–1,000
1,000–1,500
1,500–2,000
2,000–2,500
>2,500	65
21
9
2
2
1	

To model the spatial spread and extent of livestock depredation, we used the presence or absence of conflict as the response variable. We used a generalized linear model (GLM) that included 14 continuous variables as predictors (explanatory variables) with the binomial distribution respectively using the package ‘Desctools’ (Signorell et al., 2019) and ‘manipulate’ (Racine, 2012). For binomial distribution, the presence/absence of the conflict incidents (response variable) was recorded on a binary coding basis (1 = presence of a conflict incident in a grid, 0 = absence of a conflict incident in a grid). The absence of conflict was taken as the reference variable when running the models. We selected the GLM due to its suitability for modeling binary response data (Fernandes et al., 2021) and its balance between simplicity and interpretability, aligning well with our research objectives. All the variables were standardized using Z transformation, ensuring that they were on the same scale for easier comparison of their contributions in the GLM. Before model construction, a multi-collinearity test was done for all the variables based on variance inflation factor (VIF) functions using the package ‘faraway’ (Boomsma, 2014). Since none of the variables showed any multi-collinearity (VIF value > 5), we used all the variables for model construction (Chatterjee & Hadi, 2013).

To select the most parsimonious model among the set of models, Akaike’s Information Criterion (AIC) was performed (Akaike, 1973). Using the function ‘dredge’ under the package ‘MuMIn’ (Barton, 2009), all possible models were constructed and ranked based on small-sampled AICc (Barton & Barton, 2020). The final model was obtained by averaging the top candidate models (delta AIC ≤ 2) (Burnham & Anderson, 2001). To test the predictive performance of the dominant model, we generated the receiver operating characteristic (ROC) curve and area under the curve (AUC) value using the package ‘ROCR’ (Sing et al., 2005). We obtained high model uncertainty among the selected predictors indicated by similar model weight. Thus, we did full model averaging (Table 3 and Table 4) to compute the effect of predictors. Additionally, we created an odds ratio plot to visualize the effect sizes of the predictors. The plot displayed the odds ratios with 95% confidence intervals, aiding in the interpretation of predictor variable impacts on the presence or absence of conflict incidents using the package ‘ggplot 2’ (Wickham, 2016).

Table 3 GLM model with the binomial structure for the probability of livestock predation by leopard.

	Estimate	Std. Error	z value	Pr(>—z—)	
(Intercept)	−2.75313	0.541284	−5.0863	3.65E−07	
LR	0.243474	0.213269	1.141631	0.2536	
RL	0.334842	0.163479	2.048231	0.0405	
Slope	−0.59343	0.234312	−2.53266	0.0113	
DPA	−0.37682	0.192787	−1.9546	0.0486	
Elevation	1.144401	0.5124	2.233416	0.0255	
DS	−1.16784	0.468449	−2.493	0.0127	
DR	−0.3269	0.534292	−0.61183	0.5407	
DWB	−0.40473	0.667943	−0.60594	0.5446	
AF	0.503323	0.263548	1.909795	0.0462	
AGL	−0.13252	0.247977	−0.53439	0.5931	
AAL	0.333532	0.146614	2.274901	0.0229	
ASL	−0.15425	0.235238	−0.6557	0.5120	
ABG	−1.52401	1.225288	−1.24379	0.2136	
AWB	0.100116	0.154925	0.646221	0.5181	

Table 4 Second order Akaike Information criterion scores (AICc, ΔAIC & AIC weight) of a generalized linear model with binomial structure.

Component models*	df	AICc	ΔAIC	AIC weight	LogLik	
Binomial distribution						
AAL+ABG+AF+DPA+DS+Elevation+RL+Slope	9	−162.742	343.9	0.00	0.018	
AAL+ABG+AF+DPA+DS+RL+Slope	8	−164.005	344.3	0.44	0.014	
AAL+ABG+AF+DPA+DS+Elevation+RL+LR+Slope	10	−162.168	344.8	0.94	0.011	
AAL+ABG+AF+DPA+DR+DS+Elevation+RL+Slope	7	−165.405	345.1	1.17	0.010	
AAL+AF+DPA+DR+DS+Elevation+RL+Slope	10	−162.433	345.4	1.47	0.008	
AAL+ABG+AF+AWB+DPA+DS+Elevation+RL+Slope	10	−162.460	345.4	1.53	0.008	
AAL+ABG+AF+AGL+DPA+DS+DWB+Elevation+RL+Slope	10	−162.570	345.6	1.75	0.007	
AAL+ABG+AF+ASL+DPA+DS+Elevation+RL+Slope	10	−162.599	345.7	1.80	0.007	
AAL+ABG+AF+AGL+DPA+DS+Elevation+RL+Slope	10	−162.646	345.8	1.90	0.007	

We generated the conflict probability map through the maximum entropy (MaxEnt) modelling. MaxEnt is a widely used approach (Fitzpatrick, Gotelli & Ellison, 2013; Phillips, Anderson & Schapire, 2006), demonstrating the best predictive power across all sample sizes (Elith et al., 2011; Wisz et al., 2008). The occurrence points of conflict were filtered by keeping at least 100 m of the distance between locations, thereby minimizing spatial autocorrelation (Karki & Panthi, 2021; Adhikari et al., 2022). For this purpose, the georeferenced points were spatially thinned using the SpThin package (Aiello-Lammens et al., 2015). Using this technique, we utilized the spatially filtered conflict occurrence points alongside the variables with high significance in the GLM models to predict the conflict probability in the study area (Adhikari et al., 2022). The MaxEnt program (version 3.4.4) was set up to use 70% of the data points for training and 30% for model validation. A maximum iteration limit of 1,000 was chosen, and models were replicated ten times to generate average model information (Barbet-Massin et al., 2012). We assessed the model accuracy through the threshold-independent (AUC-ROC) method and the threshold-dependent true skill statistics (TSS) method. AUC value ranges from 0 to 1. Similarly, TSS ranges from −1 to +1, with the model with a higher value in both representing good predictive performance (Allouche, Tsoar & Kadmon, 2006). The continuous probability map obtained from MaxEnt was reclassified in ArcGis 10.5 to obtain a distinct probability of conflict (low = <0.40, medium = 0.4–0.6, and high = >0.6).

Results

Seasonal and temporal patterns of livestock predation

Livestock losses to leopards varied significantly with respect to season (χ2 = 15.738, df = 2, p-value < 0.001) and month (χ2 = 28.665, df = 11, p-value < 0.002). Winter exhibited the highest average count of approximately 37.00, suggesting a potential peak in livestock depredation during this season. Summer followed closely with an average count of approximately 34.33, while monsoon had the lowest average count of about 20.33. The standard errors for all three seasons were relatively consistent, indicating a similar level of variability in the counts within each season (Fig. 2). Similarly, there was also a significant difference in the timing of leopard attacks on livestock (χ2 = 56.745, df = 5, p-value < 0.001). The “4 PM to 8 PM” consistently recorded the highest average count, averaging approximately 30.33 livestock depredation, signifying a recurrent peak in the phenomenon during the late afternoon and early evening hours. Following closely, the “8 PM to 12 AM” exhibited a substantial average count of approximately 14.33, indicating continued activity during the evening and early night.

Figure 2 Mean count for (A) month, (B) season and (C) with error bars in the study area during 2017–2020.

Conversely, the “12 PM to 4 PM” displayed a significantly lower average livestock depredation count, with an average of approximately 13.33. The time period between “8 AM to 12 PM” had a moderate average count of about 12.33, while the “12 AM to 4 AM” and “4 AM to 8 AM” had the lowest average counts, each hovering around 12.00, suggesting relatively low activity during the late night and early morning. The standard error values for each time period indicate the precision of the mean estimates and the extent to which the counts may vary within each specific time period (Fig. 2).

Livestock losses and characteristics of kill sites

During the three years, 439 livestock were reportedly killed by leopards in three districts. Goats (Capra aegagrus hircus) were the most predated livestock (66.5%), followed by sheep (Ovis aries) (22.5%), cows/oxen (Bos taurus) (7%), and buffalo (Bubalus bubalis) (4%). The analysis of data across slope and elevation revealed that about 80% of kill sites occurred within a 30-degree slope, and about 90% occurred within 3,000 m elevation. In inside conservation areas, there was a lower percentage of livestock depredation (41%) than in outside conservation areas (59%). It is apparent from the kill sites that about 65% of livestock depredation in the study area was recorded within 1 km of a road. The number of livestock depredation events decreased with distance from roads. Similar patterns were observed when conflict locations were analyzed in relation to distance from rivers and settlements. About 70% of livestock depredation was observed within 1 km of a river and 85% within 1 km of a settlement (Table 2).

Influence of variables on leopard attack on livestock

In our study, we examined the significance of various variables regarding the presence or absence of livestock depredation by leopards. River length exhibited a significant positive association with livestock depredation (ß = 0.3348; p = 0.0405), indicating that areas with longer rivers were more susceptible to leopard attacks on livestock. Additionally, slope displayed a significant negative relation with livestock depredation (ß = −0.5934; p = 0.0113), suggesting that areas characterized by steeper terrain experienced fewer instances of leopard predation. The proximity to the protected area demonstrated a significant negative correlation with livestock depredation (ß = −0.3768; p = 0.0486). This suggests that areas closer to protected areas experienced elevated risks of livestock depredation.

Moreover, elevation demonstrated a significant positive relation (ß = 1.1444; p = 0.0255), implying that higher elevations were linked to an increased likelihood of leopard attacks on livestock. Furthermore, the forest area exhibited a significant positive relation with livestock depredation (ß = 0.5033; p = 0.0462), indicating that larger forested areas heightened the risk of leopard predation. Similarly, the area of agricultural land demonstrated a significant positive association with livestock depredation (ß = 0.3335; p = 0.0229), suggesting that larger agricultural areas were more vulnerable to leopard attacks on livestock. The distance to settlements variable exhibited a statistically significant negative relationship with livestock depredation (ß = −1.1678; p = 0.0127) (Table 3). This suggests that areas closer to human settlements experienced a higher risk of leopard predation. The dominant model (GLM with binomial structure) had a receiver operating curve value of 0.85 (83.55% accuracy). We also generated an odds ratio plot to provide a visual representation of the effect sizes of our predictor variables. This plot displayed odds ratios along with their corresponding 95% confidence intervals, serving as a valuable tool for interpreting the impacts of these predictor variables on the presence or absence of conflict incidents (Fig. 3).

Figure 3 Odd ratios plot for predictor variables with 95% confidence intervals.

Among the nine different component models developed through model averaging (delta AIC ≤ 2), the results of the GLM model with binomial structure (smallest AICc = −162.742) indicate that predictors such as the area of agricultural land, bare ground, forests, distance to protected areas and settlements, elevation, length of the river, and slope appeared as the dominant model for leopard attacks on livestock in the study area (Table 4.

Probability of conflict

A total of 289.11 km2 (approximately 5% of the total study area) was considered high risk for livestock depredation by leopards. Kaski district had the largest high conflict-prone area (approximately 122.28 km2), followed by Lamjung (approximately 107.01 km2) and Myagdi (approximately 59.82 km2). Similarly, 18% of the high-risk area was incorporated within Annapurna CA (Fig. 4). The model accuracy was quite good, with an average AUC of 0.82 ± 0.09 and an average TSS 0.68 ± 0.08.

Figure 4 Leopard livestock predation risk map.

Discussion

Our study comprehensively analyzes the multifaceted interactions between leopards and livestock within our research area. By delving into various aspects of leopard predation patterns and the environmental factors that influence them, we aim to contribute significantly to understanding human-leopard conflicts and developing effective mitigation strategies.

Livestock depredation by leopards

This study revealed that leopards primarily preyed on goats, followed by sheep and cattle. This predation pattern aligns with the livestock population ratios in the study districts, as the Ministry of Agricultural and Livestock Development Nepal (MOALD, 2021) reported in 2021: 50% goats, 7% sheep, 29% cattle, and 14% buffalo. In contrast, sheep were the second most vulnerable prey livestock to leopards in the study area despite constituting the smallest proportion of the population compared to cattle and buffalo. Goats and sheep, as opposed to cattle and buffalo, exhibit less defensive behavior (Dhungana et al., 2019), making them easier prey for leopards to capture. Although the diet of the leopard ranges mainly from small to some very large prey (>100 kg) (Lovari, Ventimiglia & Minder, 2013), they generally prefer species weighing between 10–40 kg (Hayward et al., 2006) and 2–25 kg (Lovari, Ventimiglia & Minder, 2013). As a result, they exhibit similar size preferences when preying on goats and sheep. Moreover, the lesser killing of cattle and buffalos in the study area might be attributed to their large body size, which is not preferred by leopards (Lovari, Ventimiglia & Minder, 2013; Hayward et al., 2006).

Seasonal and time-based variations in livestock depredation by leopards

Our analysis showed strong variations in livestock losses among seasons, months, and times. Most leopard attacks on livestock occur during the dry season (winter and summer). Our results coincide with studies conducted by Naha et al. (2020) in the Himalayan region of India and Sijapati et al. (2021) in the Terai region of Nepal. The dry season corresponds with the planting and harvesting of agricultural crops. During these months, farmers are busy with crop harvesting and production, leaving their livestock ranging freely (Naha et al., 2020). Due to the cold temperatures in the winter, managing livestock guarding also becomes difficult for herders. Negligence by herders is regarded as one of the main contributing factors to livestock losses by predators (Maclennan et al., 2009).

Moreover, the months of the winter season (November and December) receive less rainfall and thus require free grazing of livestock in forest areas due to the limited availability of fodder for stall feeding, making them more vulnerable to leopard attacks (Dhungana et al., 2019). Several studies have reported leopard behavior as nocturnal (Odden et al., 2014; Ahmed et al., 2012; Qamar et al., 2010). However, some studies also reported higher livestock depredation by leopards during the daytime (Naha et al., 2020; Woodroffe et al., 2007). In our study, the timing of leopard attacks on livestock also varied significantly, with peak occurrence during the evening time (4 PM–8 PM), similar to the studies conducted by Dar et al. (2009) at Machiara National Park, Pakistan. Multiple factors, such as animal husbandry practices and the availability of alternate prey, influence the timing of leopard depredation on livestock (Akrim et al., 2021). To sum up, our study emphasizes the significant impact of seasonality and daily timing on leopard predation of livestock, echoing findings from various regions. These insights underscore the complex interplay of factors, including animal husbandry practices and prey availability, in shaping the patterns of leopard attacks on livestock.

Factors influencing leopard predation on livestock

This study revealed that forest area, agricultural area, slope, distance to protected areas, distance to settlements, length of the river, and elevation affect livestock kills by leopards (Table 3). In India, the availability of water resources emerged as a spatial factor influencing leopard predation rates, as previously highlighted by Karanth et al. (2013). This observation resonates with similar research conducted in arid ecosystems of Africa, where water availability has consistently been identified as a primary driver of human-carnivore conflicts (Abade et al., 2018; Beattie et al., 2020). Our study corroborates these established findings. Specifically, we found a statistically significant positive correlation between the length of the river and the probability of livestock depredation by leopards. Rural villages heavily rely on livestock rearing as a prominent profession in our study area, which is characterized by higher-sloped geographical terrain and lush forests. Previous studies, such as Rostro-García et al. (2016), have reported leopard predation on livestock in rugged areas of Bhutan. Our study revealed a contrasting pattern where the probability of livestock depredation decreased with increasing slope. This finding can be attributed to the already prevalent sloped geographical terrain in our study area, which may act as a natural deterrent for leopards, making it more difficult to access and prey upon livestock. Interestingly, the study conducted by Rather, Kumar & Khan (2020) also observed high occurrences of leopards on gentle slopes, indicating that leopard behavior and habitat preferences can vary across different regions. These findings highlight the complexity of the relationship between slope and leopard predation and emphasize the need for further research to understand better the factors influencing livestock depredation in diverse geographical settings.

With increasing distance from protected areas, we observed a decreasing trend in leopard predation on livestock. Our findings are consistent with studies conducted by Constant (2014) in South Africa. However, Naha et al. (2020) documented contrasting results and found that conflicts with leopards were higher as they moved farther away from reserves. Factors such as habitat availability, prey availability, and local community dynamics can play significant roles in shaping the patterns of leopard predation on livestock. Further research is needed to understand the underlying mechanisms driving these variations and develop context-specific strategies for mitigating human-leopard conflicts.

This study revealed that elevation was a significant factor for leopard attacks on livestock, implying that higher elevations were linked to an increased likelihood of leopard attacks on livestock. Leopards can be found throughout Nepal, from the low-lying Terai (100 m) to the Himalayan peaks (4,000 m) (Dhungana et al., 2019). This finding aligns with recent research conducted by Baral et al. (2023), which suggests that leopards are experiencing a shift in their habitat preferences in response to the effects of climate change. As temperature rises and ecosystems undergo transformations, leopards may be compelled to explore higher-elevation areas in search of suitable habitats and prey resources. Consequently, the increased presence of leopards at higher elevations may bring a higher incidence of livestock attacks in these regions.

In the rural villages inside the Annapurna Conservation Area (ACA), livestock grazing practices are prevalent throughout the year, with cattle sheds strategically positioned at different elevation ranges within the forest. This availability of prey, facilitated by the existing livestock husbandry practices, may be one of the reasons why the probability of leopard predation on livestock is positively associated with the forested area. The mid-hills region of Nepal has witnessed an increase in forest area in recent decades, which could further contribute to higher carnivore-related livestock depredations, as found by Michalski et al. (2006).

Despite this trend, studies have also shown that leopards exhibit adaptability to human-modified landscapes (Baral et al., 2021; Odden et al., 2014; Constant, Bell & Hill, 2015). A study conducted by Bista et al. (2022) in Nepal’s Kathmandu district further strengthens this assertion, revealing a substantial leopard presence in areas characterized by high human population densities. This underscores the leopard’s capacity to thrive in proximity to human settlements. Furthermore, the settlements in our study area are mostly scattered and not so densely populated, allowing leopards to dwell nearby. This adaptability, in conjunction with the easy availability of livestock near the settlements, likely contributes to the observed increase in livestock depredation with a decrease in distance from settlements.

Our findings support a positive association between livestock depredation and agricultural land, consistent with previous studies by Kshettry, Vaidyanathan & Athreya (2018) and Kabir et al. (2014), which reported higher leopard predation on livestock in human-use landscapes, especially in areas with agricultural activities. Agricultural lands provide a potential food source for leopards, including domestic livestock like cattle, buffalo, goats, and chickens. The remarkable adaptive nature of leopards (Bista et al., 2022) allows them to exploit these human-modified habitats for hunting and acquiring additional food resources. Kshettry, Vaidyanathan & Athreya (2018) even reported a higher contribution of livestock to the leopard diet from human-use landscapes. The ability of leopards to cover large home ranges and their highly adaptive nature (Abade et al., 2018) enables them to persist in areas with varying degrees of human presence (Odden, Wegge & Fredriksen, 2010). Despite their adaptive behaviors, leopards may avoid areas with higher human settlement density due to potential conflicts and increased human activity, thereby limiting their predation on livestock in such regions. Hence, our results suggest that while leopards may exploit agricultural land for hunting opportunities, they may also exhibit caution and avoid areas with denser human populations. This dual pattern intricate interplay of ecological and behavioral factors underscores the complex dynamics of human-leopard interactions, necessitating further research on spatial dynamics of settlements and their impact on human-leopard conflicts.

Conclusion

Our findings offer valuable insights for managing human-leopard conflicts in the study area and beyond. The conflict risk maps generated are instrumental in pinpointing high-risk regions where resources and policy changes can be strategically directed. Key mitigation strategies include enhancing corral structures, reinforcing stock guarding, and discouraging livestock grazing in vulnerable areas. Collaborative efforts among stakeholders are essential for successful coexistence between communities and leopards. Effective guarding is crucial in the winter and summer months, especially in December and March when leopard attacks on cattle are at their worst. Building safer or predator-proof corrals and emphasizing stall-feeding techniques, particularly in locations with forest fringes, are necessary. It is also necessary to avoid grazing livestock inside dense forests and near water bodies. If grazing is unavoidable in such areas, we advise using communally coordinated herding practices. Developing a conflict mitigation strategy that incorporates cooperation between divisional forest offices, local populations, and conservationists will be essential to encourage coexistence between people and leopards. We recommend a detailed and extensive study to understand the habitat utilization pattern of leopards at a finer scale through intensive camera trapping and radio telemetry methods in the study area.

Supplemental Information

Supplemental Information 1 Data binomial

Click here for additional data file.

Supplemental Information 2 Time and season

Click here for additional data file.

We thank Dr. Dipanjan Naha, Mr. Manoj Subedi, and Mr. Bikram Gautam for their constant support and motivation.

Additional Information and Declarations

Competing Interests

Author Contributions

Field Study Permissions

Data Availability

Bikram Shrestha is a general member of Green Governance Nepal and employee of Department of Biodiversity Research, Global Change Research Institute, Czech Academy of Sciences, Brno, Czech Republic.

Saurav Lamichhane conceived and designed the experiments, performed the experiments, prepared figures and/or tables, authored or reviewed drafts of the article, identified funding agency, and approved the final draft.

Divya Bhattarai conceived and designed the experiments, performed the experiments, prepared figures and/or tables, authored or reviewed drafts of the article, and approved the final draft.

Tek Maraseni performed the experiments, authored or reviewed drafts of the article, and approved the final draft.

Kyle J. Shaney performed the experiments, authored or reviewed drafts of the article, and approved the final draft.

Jhamak Bahadur Karki performed the experiments, prepared figures and/or tables, authored or reviewed drafts of the article, and approved the final draft.

Binaya Adhikari analyzed the data, prepared figures and/or tables, authored or reviewed drafts of the article, and approved the final draft.

Pratik Pandeya analyzed the data, authored or reviewed drafts of the article, and approved the final draft.

Bikram Shrestha analyzed the data, authored or reviewed drafts of the article, and approved the final draft.

Hari Adhikari conceived and designed the experiments, analyzed the data, prepared figures and/or tables, authored or reviewed drafts of the article, and approved the final draft.

The following information was supplied relating to field study approvals (i.e., approving body and any reference numbers):

Field experiments were approved by the Department of Forest and Watershed Management (Permission number: 279/077/078).

The following information was supplied regarding data availability:

The raw measurements are available in the Supplementary Files.

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
