# Peer review of "Landscape predictors influencing livestock depredation by leopards in and around Annapurna Conservation Area, Nepal"

_PeerJ, doi:10.7717/peerj.16516_

## Round 0.1 · original submission · Major Revisions

Dear author,

Thank you for submitting your manuscript for consideration. We have recieved three detailed reviews that are complimentary of the work while also offering constructive criticism that should be helpful in futher developing the submission.

I agree with the reviewers that while the manuscript is interesting and potentially valuable, there are a number of issues that require immediate attention. These include the justification of hypotheses, description of methods, and clarification of results. I have not included specific comments of my own as those of the reviewers are suitably comprehensive. I invite you to respond to their comments and submit your revision at your convenience.

Best wishes,
Anthony

**Language Note:** The review process has identified that the English language must be improved. PeerJ can provide language editing services - please contact us at copyediting@peerj.com for pricing (be sure to provide your manuscript number and title). Alternatively, you should make your own arrangements to improve the language quality and provide details in your response letter. – PeerJ Staff

Reviewer 1 ·

Basic reporting

Overall, this manuscript was an interesting good read. There are a few issues that do not ruin the manuscript but should be addressed by the authors.

Of the highest importance is the placement of past research and hypotheses in the manuscript. The introduction upon first reading does a good job of laying out the context of the research and seems relatively well referenced as it is written. Important potential issues arise though when comparing the introduction to the methods. The Introduction says that little work has been done on human-leopard conflict to make the case for the importance of the work reported in the manuscript. Yet the methods and discussion include a variety of other research on human-leopard conflict, including research that directly guides the hypotheses the authors intended to evaluate. It would be worthwhile to touch on the literature more directly in the Introduction to better position this work within the past research that is informing the authors’ described work. Similarly, the authors can then use the description of that past literature to lead into their hypotheses at the end of the Introduction rather than in lines 154-189. The Methods can then focus on how those hypotheses were tested (e.g., developing slope and elevation values) rather than stating part of partial methods, then stating hypotheses, and then additional methods.

The language used throughout the paper is very readable, but there are a number of instances where it should be tightened up. Some (non-exhaustive) examples:
Line 131: “we took help from Google…” vs we used Google?
Line 147/148: numbers are not in superscript
Line 195/196: list is difficult to read as winter doesn’t seem to have an i.e., but the other seasons do
Line 217: Authors define some abbreviations and not others. Recommend defining each the first time they are used. E.g., AIC defined but AICc and ROC and AUC are not but then Line 278 says “receiver operating curve (AUC) value”
Line 297/298: “population ratio of goats, sheep, cattle, and buffalo in the study districts is 50:7:29:14”. Easier to read if authors just refer to these as percentages of livestock in the area consistently (as they do in the next sentence).
Line 336: “Kabir et al. (2014) reveal” is in a present tense while the rest is in the past tense and “found” would make more sense

The Figures look pretty good overall. I think highlighting the Annapurna Conservation Area could be a bit more clear, especially in Figure 3 where the blue line is hard to see against the background at times.

Tables 1, 2, and 3 are all labelled Table 1; Table 2 is missing units on the final two distance measures (I assume they are also in meters, but still good to directly indicate).

Outside of my concern around the placement of hypotheses and the use of literature in the Introduction, the article fits the appropriate general structure and the raw data has been included.

Experimental design

I have no big picture concerns about the methods the authors report in this manuscript. I think my largest concern that the authors should address is the reliability of the depredation events and numbers they are using for their data. The authors state “only recorded species for which compensation had been allocated to livestock owners by the government.” How reliable are the numbers? Are other livestock species taken by leopards but not included in those compensation programs? Is the degree to which producers will not report losses due to administrative burden or the requirements to prove the loss known?

Other design related questions I have that the authors should address are:
- Line 132: “identify the grids of the conflict site” unsure what grids is referring to here. Coordinates? Boundaries?
- I am having difficulty finding the Raghunathan et al. 2018 article. Recommend the authors check the reference and add in some clarification on the counter-intuitive smaller grids means more respondents/grid and vice-versa
- Authors state distance from protected areas was calculated based on Euclidean distance, but where was the protected area point? Border of a protected area? Center of the protected area polygon? This would seem especially important for the livestock depredation sites within protected areas, how were those distances measured?
- Authors report statistical analyses on season, month, and time of day that are not described in Section 2.4. The authors say pivot tables for descriptive statistics but they also conduct significance testing and do not conduct follow up pair wise comparisons which would seem applicable given the number of categories.
- I am unsure why the authors did not perform any statistical testing on their landscape variables (section 3.1) but did on their temporal variables (section 3.2). Did the authors see the results for the landscape variables as uninformative due to the inclusion of the variables in the regression? If so, maybe swap sections 3.1 and 3.2 so the landscape descriptives are presented right before the regression that uses the data. Also, much of the information in section 3.1 can be gleaned from Table 2 so grouping landscape categories together for the sake of descriptives when a reader could do that via the table seems unnecessary.

Validity of the findings

I don’t have any major concerns around the validity of the findings and I think the findings can be very useful for future research and management. I think there are a few points that the conclusions could address or clarify that would help contextual the findings:
- Section 4.1: leopards prefer smaller prey and goats and sheep are also smaller and more commonly raised so more likely to be taken. But are the pastoral practices of goats and sheep also different than the pastoral practices for cattle and buffalo? Are goats and sheep more likely to be in forested areas, farther away from settlements, etc.?
- Outside of select mentions (e.g., lines 363/364), authors do not seem to mention how the distribution of leopards themselves might change across the same variables they found were associated with livestock predations. This is an important consideration and worth devoting some space to in the Discussion if the appropriate information is available.
- Lines 375-379: Authors should clarify how distance from settlements are directly related to availability of prey in forests as they state forests explain the settlement-predation relationship; e.g., is the mid-hills region most heavily populated or are authors saying something else?
- Authors mention that the conflict map they developed is important for planning and implementing mitigation measures but then discuss types of mitigation strategies (e.g., lines 405-415) rather than more direct application of their results. For instance, wouldn’t the map be useful for informing where to spend resources or where policy changes may be particularly important so how could those results be applied in a particularly high risk spot?

Additional comments

One last note, I would encourage the authors to consider whether depredation is the most correct term for their work. In many cases, leopards are taking livestock as part of the predation process with no difference to them between livestock and other wild prey. Line 67 reinforces this idea as the authors discuss leopards shifting their prey to livestock after environmental changes. With this in mind, perhaps the authors should consider using the term livestock predation rather than depredation.

Reviewer 2 ·

Basic reporting

I have reviewed the manuscript entitled “Landscape predictors influencing livestock depredation by leopards in and around Annapurna Conservation Area, Nepal”, and commend the authors on a comprehensive effort to analyze several years of monitoring data to better predict and mitigate future human-leopard conflicts. This manuscript has the potential to provide a helpful contribution to the literature on carnivore-livestock conflict and coexistence, and to address a regional data gap regarding human-leopard relations in Nepal. However, there are considerable components of the manuscript that need addressing, most seriously the interpretation and discussion of some of the key results which seem to have been erroneously interpreted. Additionally, the manuscript will benefit from clarification in several parts of the methods section (including justification of covariate choice and associated hypotheses), as well as a revision of the discussion and conclusion sections to include more discussion of broader implications of this work (beyond Nepal and beyond leopards). Please see my attached general comments and suggested edits.

Regarding the introduction and background: It would be helpful if the authors introduced more about the Annapurna Conservation Area and its conservation importance when providing background about the need for the study within the introduction. As it is, the readers leave the introduction not understanding why this study occurred in the ACA in particular, rather than anywhere else.

Experimental design

- In the Data Collection section, please describe in detail the verification process as possible. It is important to understand who and how predation data were verified in order to contextualize the broader results.
- In the Data Processing section, the authors cite a paper that makes a suggestion of grid sizes no larger than 5km2, but then go on to say they used grid sizes of 16km2, which is much larger than the recommendation they cite. It would be helpful to have some statements justifying this choice of a coarser grid size and its potential statistical ramifications.
- Please be sure to carefully think through and justify all hypotheses. For example, the “land use” subsection cites a few hypotheses about types of land uses that could be more likely to foster leopard attacks on livestock, but does not explain why. As another example, in the “human influence and infrastructure” subsection, the authors note that leopards could be more likely to prey on livestock in areas close to human settlements and roads, citing studies that show leopards inhabit locations of high human density. However, they fail to note that livestock are more likely to be found in these places as well, which would of course lead to more predation on livestock in these areas. The hypotheses regarding topography in Lines 180-182 are also not supported with clear justification. The hypothesis regarding distance to protected area has a very clear justification which is not described- citing the studies that came before is not enough.
- Please be more specific about the statistical tests used to analyze seasonality/timing of leopard attacks on livestock (i.e., in lines 196-200 and described in the results on lines 250-258).

Validity of the findings

- There are a few places in the results that need clarification or changing, and that will then impact the discussion points considerably. For example, a positive association between proximity to protected areas and livestock predation is mentioned, and the authors say that this means livestock predation is more likely to happen closer to protected areas. Yet, the variable used was “distance from protected areas”, in which case a positive relationship would mean there is more chance of livestock predation farther away from protected areas. The same issue comes with the interpretation for distance to settlements- where the authors seem to have interpreted the result in the opposite manner than it should have been. Correcting these errors will of course change the discussion considerably.
- Please include a more detailed discussion of the points you bring up in lines 329-330
- In the discussion regarding the relationship between conflict and river length (lines 334-340) you cite a study that says reduced water bodies trigger leopards to search for water, and note that the results align with that study, when in fact it seems that your results show the opposite trend (more conflict with more water availability). Please clarify and/or expand on this discussion point.
- Please contextualize your results regarding forested areas, using references to previous studies (lines 370-374).
- Please prioritize broader implications of your study in your conclusion (i.e., to human-wildlife coexistence endeavors), rather than just focusing on local or leopard-specific implications.

Additional comments

Line and grammatical edits:
- Suggested edit for Lines 26-29: “In this study, we investigated predictors associated with livestock depredation by leopards using 274 cases of leopard attacks on livestock that occurred between 2017 and 2020 in the Annapurna Conservation Area, Nepal.
- Line 30: change to “Of 14 predictors analyzed, results revealed that forest, agricultural land, length of rivers, etc… significantly predicted the probability of leopard attacks on livestock”
- Throughout: consider using either “predation” or “depredation”, but not both
- Include Latin/scientific names when species are first mentioned in the abstract, and again the first time they are mentioned in the body of the manuscript
- Line 34: change to “depending on the livestock species”
- Line 37: It is unclear how this conclusion is similar to the previous one, so you can probably remove “Similarly,”
- Line 42: Include additional keywords in your list, such as leopard and carnivore-livestock conflict
- Line 47: change to “…and wild carnivores can lead to human-carnivore conflicts”
- Line 51: replace “reasons for the” with “types of”
- Line 59: this Latin name should have been introduced in the previous paragraph where leopards are mentioned
- Line 64: Change to “Threats to leopards include habitat degradation…”
- Lines 59-69: This paragraph is a bit repetitive. Rather than repeating, consider including some additional depth about the types of challenges that are created in regard to people’s livelihoods and safety when sharing landscapes with leopards
- Line 77: change to “The effect of livestock depredation is particularly severe…”
- Line 79-80: It’s a bit sudden to introduce the Annapurna Conservation Area in this paragraph, rather than in the next one. Think of the introduction starting broad and then increasingly narrowing in. The beginning of the following paragraph is still broad, and comes to the ACA later.
- Line 88: Please cite this statement about leopards being one of the least studied carnivores in Nepal, or back it up with a statement about why that could be the case. Same with the following sentence about landscape variables.
- Line 91-94: Make sure you maintain a consistent tense (present or past, not both)
- Line 105: it says “128 wild animals” but then goes on to list hundreds of species of wild animals. If you mean mammals, please say so. Similarly, please change this sentence to say “128 mammal species, 514 bird species, 340 butterfly species…” etc. if species is what you are referring to
- Line 108: change “owns” to “own”
- Line 110-111: move the phrase at the end of the sentence to the beginning of the sentence, i.e., “To ease its management, ACA is divided into seven-unit conservation offices:”
- Line 111: change to “The local community members reside in”
- Line 112: change to “This study was conducted in three of the five Gandaki Province districts in which people reside (Kaski, Lamjhung, and Myagdi), which overlap with the ACA.”
- Line 121: change to “including attacks by the common leopard”. change next sentence to “We compiled compensation data for leopard attacks on livestock…”
- Line 128: DFO was not defined prior to this sentence. Please define all acronyms
- Line 147-148: I believe you mean 4km x 4km (not 4km2).
- Line 176: change to “the length of roads in each cell”
- Line 179: please be clearer about what you mean by “different factors”
- Line 196: “data” should not be capitalized
- Line 199: I think a word is missing here- is it “daily”?
- Line 201-202: change to “we used presence or absence of conflict as the response variable” – make sure to explicitly state that you used absence of conflict as the reference variable when running the models
- Line 217: include apostrophes around ROCR as per the rest of your notation regarding packages
- Line 224-226: please clarify the data filtering process
- For all results cited in lines 261-279, please include the estimate. Also, p values can be cited as p<0.05, without an asterisk.
- Line 263: do you mean “longer” rather than “larger”?
- Line 278-279: this sentence seems out of place. Please remove
- Line 319: change “tough” to “difficult”
- Line 334: remove “an”
- Table 2 is included in the manuscript as a second Table 1, which seems to be a mistake. Also, for any distance measurements, be sure to include the unit (as in, are they all in meters? Make that clear).
- Table 3 is also included in the manuscript as “Table 1” – please fix. Also explicitly write out “generalized linear model”

Reviewer 3 ·

Basic reporting

English is mostly good. References are really lacking/too few.

Structure is mostly good, and data is shared.

Experimental design

See "Additional comments" section - I have issues here that need addressing.

Validity of the findings

Findings appear valid but can be improved in terms of presentation and require more clarity in terms of methods.

Additional comments

The authors assess livestock depredation by leopards in Annapurna Conservation Area, Nepal, and describe how livestock depredation varies by species, season, and time. They have included their data and research permit/permission along with the manuscript file.

Overall, the manuscript is of moderate quality, and will require major revisions to be considered for publication, in my opinion. Comments below.

General comments:

More detail is needed on the statistical analyses done in the methods section – I urge the authors to include an expanded data analysis sub-section where they explicitly tell the reader what alpha/significance cutoff was used (I assume .05?), more details on the MaxEnt approach used and how it works, WHY the different predictor variables were scaled/transformed (I see you did a Z transformation, but why?), and why they chose a GLM when a GLMM or hierarchical approach might/likely would have been a more appropriate choice. Science needs to be reproducible, and at the moment, the study falls short. This is not meant to discourage the authors, but to urge them to embrace the spirit of scientific ethics and rigor. It’s a common issue I find as a reviewer, and the authors here absolutely did an above-average job of describing their quantitative approach, but it can be refined and clarified more.

Similarly, for the results, I would have appreciated some plots showing the actual EFFECT SIZES of the predictions rather than the very dense model tables that were at the end of the manuscript. There are many packages that can be used in combination with {ggplot2} to extract quantified predictions (with confidence intervals) to plot predictions.

Format table 1 so that the entire contents of each cell are on one line, if possible.

The authors explain and rationalize their general decisions quite well, and I commend them for that.

Please use less passive voice – e.g., in line 30 you say “fourteen potential predictors were analyzed” which sounds a bit clumsy. The manuscript suffers throughout from heavy use of passive voice.

In general, the writing style is a bit odd, and I find too many fancy/flowery words are used in slightly strange ways. I suggest a close re-read and revision to improve clarity and directness.

The introduction and discussion can use more of a logical flow, i.e., going from general to specific for the introduction and from specific to general in the discussion. This sort of “hourglass shape” of writing helps the reader digest and interpret your story and results best.

Please make figures colorblind-friendly.

Keywords: please add more. 3 is very little.

I found the references/works cited to be underwhelming and believe the manuscript would improve considerably if the authors referenced the wider carnivore-livestock literature more thoroughly. It does not exploit the wider field enough to embed the study well. Some suggested papers below:

https://www.sciencedirect.com/science/article/pii/S0006320722000684

https://peerj.com/articles/7916/

https://onlinelibrary.wiley.com/doi/abs/10.1111/j.1749-4877.2012.00303.x

https://www.frontiersin.org/articles/10.3389/fcosc.2021.691975/full

https://journals.plos.org/plosone/article?id=10.1371/journal.pone.0162685

https://besjournals.onlinelibrary.wiley.com/doi/10.1111/1365-2656.13812

https://onlinelibrary.wiley.com/doi/abs/10.1002/ece3.3565

https://onlinelibrary.wiley.com/doi/pdf/10.1002/9781444317091#page=149

In the abstract, instead of just saying you “analyzed” the effects of the various predictors, it would be good if you explained briefly HOW you did it, e.g., if you used a GLMM or some other type of model. Of course, a lot of detail is not needed in an abstract, but it would be helpful if you at least said what you did.

Also – how did the authors know that leopards were the predators? Was this an assumption they made? How did local community members know it was not another extant predator – e.g., one of the extant bear species, striped hyenas, tigers, or snow leopards?

Moreover, the entire discussion section was the weakest part of the manuscript and did not feel like a discussion at all. The authors report too many numbers here at the start and it feels like a messy blend between an introduction and results section. They also went into way too much detail regarding every single variable, rather than focusing on the 3-4 main findings and the significance of the findings for human-carnivore coexistence. Please change this entire section; it will require a thorough re-write and can be shortened considerably.

Specific comments:

Line 31: you have merged the methods and results in the same sentence here, which is bizarre. Please split this into two sentences.

Lines 31-38: the way the results are reported in the abstract is a bit odd to me. You mention length of rivers, slope etc. at the start, then begin talking about species and season and time, then go back to describing the effect of slope, elevation etc. in a bit more detail... it feels jumbled and does not read well. Please re-organize this bit.

Line 32: slope of what? Too vague.

Line 45: replace “emerges” with “has emerged”.

Line 55: when mentioning leopards here, you need to add their scientific name Panthera pardus when used for the first time. You only use it the second time in the next paragraph.

Lines 65-66: it would be useful to cite and mention their IUCN status here.

Line 68: you go from discussing leopards (specific) to then talking about “predators” (general) again. I suggest removing this sentence entirely.

Line 81: again, now you are going back to very general abstractions about human-carnivore coexistence. A more logical flow is to go from general in the first 1-2 paragraphs, then get more specific about leopards, Nepal, Annapurna etc. later on in the introduction.

Line 88: is this really true, relative to e.g., striped hyenas?

Line 111: it would be good to mention what ethnic groups/tribes these “locals” comprise of or come from and some more information on their general lifestyle. I know you said they are pastoralists, but do they engage in e.g., subsistence agriculture or other activities?

Figure 1: the red color used for grass and the conservation area boundary is too similar. Please contrast them more or use a different of colors/textures.

Line 121: I find it a bit odd that you keep calling it the “common leopard” after earlier saying you’d just call it the “leopard”; please be consistent

Line 121: you need a citation at the end of this first sentence

Line 147: please use a superscript for the 2 here so it becomes “squared”.

Line 154: I find the use of bold with a colon here for the subheading a bit odd because it does not follow the formatting used over the rest of the manuscript. I suggest using something more intuitive, e.g., just italicize with indent for terms such as “Land use” to elaborate on the different variables you tested.

Line 161: which version of ArcGIS?

Line 191: since you are using a temporal predictor, I wonder why you do not include test(s) for temporal autocorrelation (see the package ‘spaMM’) or some sort of time-series analyses.

Line 239: add scientific names for the different livestock species.

Line 239-248: it would be good here to say how many of the grids actually were scored “1” given that you used a binomial distribution. If it was mostly 0s, then you may need to use a zero-inflated binomial (or even beta) regression. Please report and check.

Line 242-244: it’s not entirely clear what you are saying here. You’re saying that 51% of the kills occurred within conservation areas? If so, just say that – you phrased it weirdly.

Lines 251-254: please explain why you reported and describe these percentages as “pooled” totals rather than means with standard error bars. You collected your data over a 3-year period, so it seems odd to pool all of it together.

Lines 261-279: I have a major issue here. You are only reporting p-values here, but it would be better if you also reported effect sizes in a way for the reader to understand and interpret the results. For instance, for every extra km of river length, how much does predicted livestock depredation increase? Merely reporting the p-value is insufficient – you can easily plot your predictions and show the actual predicted effects using packages such as {ggeffects} or other similar options. I also do not understand why you are adding an asterisk to the p-values; if you just mention in the methods that an alpha of 0.05 was deemed significant, you can remove the asterisks.

Line 290: “quite good” seems very subjective.

Lines 295-298: why are you reporting results in the discussion? Move this!

Line 303: the first paragraph of a discussion should be a general synthesis of the study and summary of what the study covered/did. You jump right into recapping results and even introducing new things, which makes for a weak start to the discussion.

Lines 303-306: why only bring up prey preference now? This should be in the introduction where you introduce leopard natural history/ecology.

Lines 311-330: this feels more in line with what a discussion section should entail, but the paragraph lacks a strong concluding sentence.

Lines 336-337: and where did the Kabir et al. study take place?

Line 340: I find the random italicization of variables in the discussion section to be very distracting. Just use normal formatting please.

Lines 332-397: you have an exhaustive set of paragraphs here that go through each variable one at a time, which is entirely unnecessary and tiring to read. Just focus on the key, fundamental findings of your study and the most information that managers can use from it to promote human-carnivore coexistence. I also am surprised not to see more discussion about the wider field of human-livestock interactions well beyond Nepal and South Asia in general. You spend way too much time here focused on your findings rather than the wider significance and importance of your study, and how it advances science.

Reference list ideally should have DOIs added. In general, it is short and could use a more thorough literature review and search effort.

---

## Round 0.2 · accepted · Accept

Dear author,

I approached the prior reviewers to request their feedback on your revision. Unfortunately, only one reviewer was available to review the revised manuscript. However, the singular reviewer has recommended acceptance. I have reviewed the submission myself and concur. You have comprehensively addressed the reviewer's comments and I believe that the revised manuscript is now ready for publication. Congratulations.

Best wishes,
Anthony

Reviewer 3 ·

Basic reporting

The authors thoroughly improved the writing and breadth/depth of the literature review. Raw data is available; results are relevant to hypotheses and study design and concept.

Experimental design

The statistical framework is more robust and any issues I had were well-explained and addressed in the rebuttal.

Validity of the findings

Looks sound after having checked the data and analyses.

Additional comments

Good job addressing a lot of comments from three different reviewers. Happy to recommend acceptance.